# Effect of methylene blue on experimental postoperative adhesion: A systematic review and meta-analysis

**Su Hyun Seo**, **Geun Joo Choi** [ORCID], **Oh Haeng Lee**, **Hyun Kang** [ORCID]*

Department of Anesthesiology and Pain Medicine, Chung-Ang University College of Medicine, Seoul, Republic of Korea

These authors contributed equally to this work.
* roman00@naver.com

**Data Availability Statement:** All relevant data are within the paper and its Supporting information files.

**Funding:** The author(s) received no specific funding for this work.

## Abstract

Adhesion is a primary challenge following surgery, and the anti-adhesive effect of methylene blue (MB) has been investigated. This systematic review and meta-analysis aimed to evaluate the effect of MB on postoperative adhesions in experimental studies. We initially searched OVID-MEDLINE, EMBASE, and Google Scholar in February 2021, and then in May 2021. The anti-adhesive efficacy of MB was compared with that of the control (either placebo or nothing) after the surgical procedure. The primary and secondary outcomes were the macroscopic and microscopic adhesion scores, respectively. Traditional meta-analysis, meta-regression, and trial sequential analysis (TSA) were performed to analyze the retrieved outcomes. We included 13 experimental studies of 367 rats (200 rats received MB and 167 rats received placebo or nothing). The macroscopic adhesion scores were significantly lower in the MB-administered group than in the control group (standardized mean difference, 2.313; 95% confidence interval, 1.104 to 3.523; $I^2$ = 94.0%, Tau = 2.059). Meta-regression analysis showed that macroscopic adhesion tended to decrease with an increase in MB dose. TSA demonstrated that the cumulative $Z$ curve crossed both the conventional test and trial sequential monitoring boundary for the macroscopic adhesion score. MB had a beneficial effect on intraperitoneal adhesion following laparotomy, and adhesions decreased with increase in dose.

## Introduction

Postoperative adhesion is a natural healing and repair process after surgery [1]. However, postoperative adhesion causes various complications, such as bowel obstruction, female infertility, difficulty in reoperation, and chronic pain after surgery [2, 3], resulting in an increase in readmission rate, hospital stay, and medical expenses after surgery [4–7].

Therefore, various strategies, including minimal traumatic manipulation [8]; frequent irrigation [8]; placing mechanical barriers on the surface of damaged tissue, such as film type [1, 9, 10], solution [1, 11], and gel-type [1, 11]; applying chemical barriers, such as statin [12],

**Competing interests:** The authors have declared that no competing interests exist.

non-steroidal anti-inflammatory agents, heparin [13], fibrinolytic agents [14], thrombin-activated fibrinolysis inhibitors [15], and a combination of mechanical and chemical barriers [16], have been developed and employed to prevent postoperative adhesion. However, it remains a major challenge following surgery and is not completely and consistently controlled [10].

Methylene blue (MB), commonly used as a medical dye, has some theoretical potential to prevent postoperative adhesion: 1) anti-oxidant effect by inhibiting the production of oxygen radicals [17], 2) anti-bacterial effects, and 3) anti-inflammatory effects by inhibiting interleukins (IL-1, IL-6) and tumor necrosis factor-alpha [18].

The anti-oxidant properties may also be presented by blocking electron transfer through the xanthine oxidase effect [19, 20], which may prevent or suppress adhesion or enhance the fibrinolytic system. The antibacterial properties of MB suppress bacterial infections and accelerate recovery, which in turn prevent postoperative adhesion, since if the wound healing process (an inflammatory response) is prolonged, adhesions can easily occur. Thus, the anti-inflammatory effect of MB can also inhibit the formation of adhesions.

To take advantage of these properties, many studies have investigated the anti-adhesive effect of MB; however, the results have been inconsistent [17, 21–23]. Furthermore, there are currently no systematic reviews or meta-analyses investigating the effect of MB on adhesion formation after surgery.

Therefore, this systematic review, meta-analysis, meta-regression, and TSA aimed to critically review and summarize the currently available evidence from experimental studies investigating the efficacy of MB in terms of postoperative adhesion.

## Methods

### Protocol and registration

We developed the protocol for this systematic review and meta-analysis in accordance with the preferred reporting requirements for systematic review and meta-analysis protocol (PRISMA-P) statement, and registered the protocol in the PROSPERO network (registration number: CRD42021211602; www.crd.york.ac.uk/Prospero) on February 04, 2021.

This systematic review and meta-analysis on the effect of MB on experimental postoperative adhesion was performed according to the protocol recommended by the Cochrane Collaboration [24], and reported according to the guidelines of the PRISMA [25]. The methodology for this systematic review and meta-analysis was based on a previous study [12].

### Eligibility criteria

The inclusion and exclusion criteria for this study were determined before conducting the systematic search. All animal studies that compared the effects of applying MB to the surgical site with that of a control, for the prevention of postoperative adhesion, were included. Review articles, case reports, case series, letters to the editor, commentaries, proceedings, laboratory science studies, and other non-relevant studies were excluded.

### Literature search

Two authors (Seo SH and Choi GJ) independently carried out database searches using OVID-MEDLINE, EMBASE, and Google Scholar in February 2021, and then in May 2021. The search strategy, which included a combination of free text, Medical Subject Headings, and EMTREE terms, is described in the *Appendix* in S1 File. Reference lists were imported into Endnote software 9.3 (Thompson Reuters, CA, USA) and duplicate articles were removed. To identify all relevant articles, we scanned the reference lists of the selected original papers until

no further relevant references could be found. No language or date restrictions were imposed. We planned to consult and co-work with experts affiliated with our university for foreign language translation, when needed.

## Study selection

The titles and abstracts identified through the search strategy described above were reviewed independently by two investigators (Choi GJ and Lee OH). If a report was determined to be eligible from the title or abstract, the full paper was retrieved. Potentially relevant studies chosen by at least one author were retrieved, and full-text versions were evaluated. To minimize data duplication due to multiple reports, papers from the same authors, organizations, or countries were compared. Articles that met the inclusion criteria were assessed separately by two investigators (Choi GJ and Lee OH), and any disagreements were resolved through discussion. In cases where an agreement could not be reached, disputes were resolved with the help of a third investigator (Kang H).

## Data extraction

All interrelated data from the included studies were independently extracted and entered into standardized forms by two authors (Choi GJ and Seo SH), and then cross-checked. When authors disagreed, the article was re-evaluated by each author until a consensus was reached. If no consensus was reached, a third investigator (Kang H) was consulted.

We treated MB administration at the surgical site as the MB group regardless of type, dose, or administration method, and treated those with placebo (saline) and nothing administered at the surgical site as the control group. If a study reported outcomes for both placebo and nothing, we combined both results for the analysis of the overall effect of MB, and separately performed a subgroup analysis for the placebo or nothing as control. We also combined all MB groups if a given study had more than one MB group that was eligible for comparison for the analysis of the overall effect of MB.

The standardized form included the following items: (1) title, (2) name of first author, (3) name of journal, (4) year of publication, (5) types of animal studied, (6) type of surgery performed, (7) interventions in control group, (8) interventions in experimental group, (9) definition of macroscopic adhesion score, (10) definition of microscopic adhesion score, (11) severity and extent of macroscopic adhesion, and (12) severity and extent of microscopic adhesion score.

The data were initially extracted from tables or text. In cases involving missing or incomplete data, an attempt was made to contact the study authors to obtain relevant information. Some data were presented as figures rather than numbers [26–29], and the open-source software Plot Digitizer (version 2.6.8; http://plotdigitizer. sourceforge.net) was used to extract the numbers.

## Methodological quality and publication bias

The methodological quality of the selected studies was assessed for five domains: (1) random allocation into treatment and control groups, (2) husbandry conditions (light/dark cycle, temperature, access to water, and environmental enrichment), (3) compliance with animal welfare regulations, (4) potential conflicts of interests, and (5) whether the study appeared in a peer-reviewed publication. Two authors (Choi GJ and Seo SH) independently evaluated the methodological quality of the studies, and any discrepancies were resolved by a third investigator (Kang H).

## Outcome measure

We recorded outcomes according to intention-to-treat analysis, where available. The primary outcome measure of this meta-analysis was the severity of adhesion under macroscopic evaluation (gross adhesion score). The secondary outcome measure was the severity of microscopic adhesion scores. In addition, the side effects of MB treatment were evaluated in this systematic review and meta-analysis.

## Statistical analyses

*Ad hoc* tables were created to summarize data from the included studies by listing their key characteristics and any important questions related to the review objectives. After extracting relevant data, the investigators determined the feasibility of the meta-analysis. Two authors (Choi GJ and Kang H) independently input all the data into the software. The standardized mean differences (SMDs) and their 95% confidence intervals (CIs) were calculated for each outcome. Between-study heterogeneity was assessed using Cochran's Q and Higgins's $I^2$ statistics. A P-value of $< 0.10$ for the $chi^2$ statistics or an $I^2$ greater than 50% was considered to indicate heterogeneity. When the combined data that showed heterogeneity was less than 10, t-statistics (Hartung-Knapp-Sidik-Jonkman method) was used instead of the Z-test in all analyses to lower the error rate [30].

Subgroup analysis was conducted according to the type of control group (saline vs. nothing) and surgical procedure (uterine horn vs. cecum or colon). We also conducted sensitivity analyses to evaluate the influence of individual studies on the overall effect estimate by excluding one study at a time from the analysis. If the reported data were medians (range, $P_{25}$–$P_{75}$), medians (ranges), or means (standard error of means), means and standard deviations were calculated from these values [31].

Funnel plots were drawn for each data as a measure of publication bias across studies, which were assessed visually for symmetry. Considering the small study effect, we also estimated publication bias using Egger's linear regression test. If the funnel plot was asymmetrical or the P-value was found to be $< 0.1$ by Egger's test, the presence of a publication bias was considered, and we conducted a trim and fill adjusted analysis to remove the most extreme small studies from the positive side of the funnel plot. We then recalculated the pooled dropout prevalence at each iteration until the funnel plot was symmetric to the (new) pooled dropout prevalence [32]. When fewer than 10 studies were included, publication bias was not estimated.

To evaluate the association between macroscopic adhesion score and MB dose, univariate meta-regression was conducted. In the meta-regression analysis, the doses of MB in each arm were the independent variables, and macroscopic adhesion score was the dependent variable.

## Trial sequential analysis

We performed a trial sequential analysis (TSA) on the macroscopic adhesion score to calculate the required information size (RIS), and we assessed whether our results were conclusive. We used a random-effects model to construct the cumulative Z-curve. TSA was performed to maintain a 5% overall risk of type I error. If the cumulative Z-curve crossed the trial sequential monitoring boundary or entered the futility area, a sufficient level of evidence to accept or reject the anticipated intervention effect may have been reached, and no further studies were needed. However, if the Z-curve did not cross any boundaries and the RIS was not reached, the evidence to conclude was insufficient, indicating the need for further studies [33].

For the macroscopic adhesion score, we used the observed standard deviation (SD) in the TSA, a mean difference of the observed SD/3, an alpha of 5% for all outcomes, a beta of 10%, and the observed diversity as suggested by the trials in the meta-analysis.

We performed analyses using comprehensive meta-analysis software (version 2.0, Biostat, Englewood, NJ, USA) for traditional meta-analysis, meta-regression analysis, and TSA software (Copenhagen Trial Unit, Centre for Clinical Intervention Research, Denmark) for the sequential trial analysis.

## Results

### Study selection

From OVID-MEDLINE, EMBASE, and Google Scholar, 42 studies were initially identified, and a subsequent manual search revealed four additional studies. After adjusting for duplicates, 41 studies remained. Of these, 22 studies were discarded after reviewing their titles and abstracts. The remaining 19 studies were reviewed in detail, after which five studies were excluded for the following reasons: 1) they were human studies [34] and 2) they did not report the outcomes of interest [20, 35–37] (Fig 1). All studies reviewed in full text version were written in English.

The kappa value for the selected articles between the two reviewers was 0.826.

### Study characteristics

The characteristics of the included studies are summarized in Table 1.

The types of surgeries performed included laparotomy of the cecum [21, 38, 39], uterine horns [22, 28, 29, 40–42], colon [17], and unspecified laparotomy types [23, 27, 43, 44]. Male Wistar rats [27, 39], female Wistar rats [21, 28, 38, 40, 41, 43], female Wistar albino rats [22, 29], Wistar albino rats (sex not specified) [42], male Sprague-Dawley rats [17], and female Sprague-Dawley rats [23, 44] were used. For the control group, normal saline was used as a control [27, 28, 38, 39, 41] and both sham and normal saline [17, 21, 22, 29, 40, 42–44]. For the experimental group, the concentrations of MB were 1.0% [17, 21–23, 28, 29, 41–44], 0.525% [38], and 30 mg/kg [27]. For the experimental group, Kluger et al. used 0.13%, 0.25%, 0.5%, and 1.0% of MB with adhesion induction and 1.0% of MB without adhesion induction [40], whereas Mahdy et al. used 0.5%, 1.0%, 5.0%, and 9.0% of MB [39].

### Macroscopic adhesion score

Thirteen studies (including 367 animals) measured the macroscopic adhesion score.

Macroscopic adhesion scores were reported based on a 5-point scale [17, 21, 23, 38–41], 4-point scale [22, 28, 42], 5- and 6-point scale [29], 14-point cumulative scale [43], or the percentage of ischemic buttons with fibrinous postoperative adhesions [27].

The effect of MB was compared with that of saline in 12 studies, nothing in one study [42], and saline and nothing in four studies [17, 21–23]. Thus, we compared the effect of MB with that of saline, nothing, and a combination of saline and/or sham (Table 2).

When macroscopic adhesion scores were compared with the combined results of using saline and nothing as control, the macroscopic adhesion score was significantly lower in the MB group (SMD, 2.313; 95% CI, 1.104 to 3.523; $I^2$ = 94.0%, Tau = 2.059) (Fig 2).

There was no change in the significance of the results after performing a sensitivity analysis by removing one study at a time (Fig 3).

Subgroup analysis based on surgical procedures showed that the macroscopic adhesion score was significantly lower in the MB group in laparotomy of the uterine horn (SMD, 1.990; 95% CI, 0.100 to 3.881; $I^2$ = 94.1%, Tau = 2.075); however, there was no evidence of differences between groups in laparotomy of the cecum or colon (SMD: 2.389; 95% CI, –1.075 to 5.852; $I^2$ = 96.89%, Tau = 3.465).

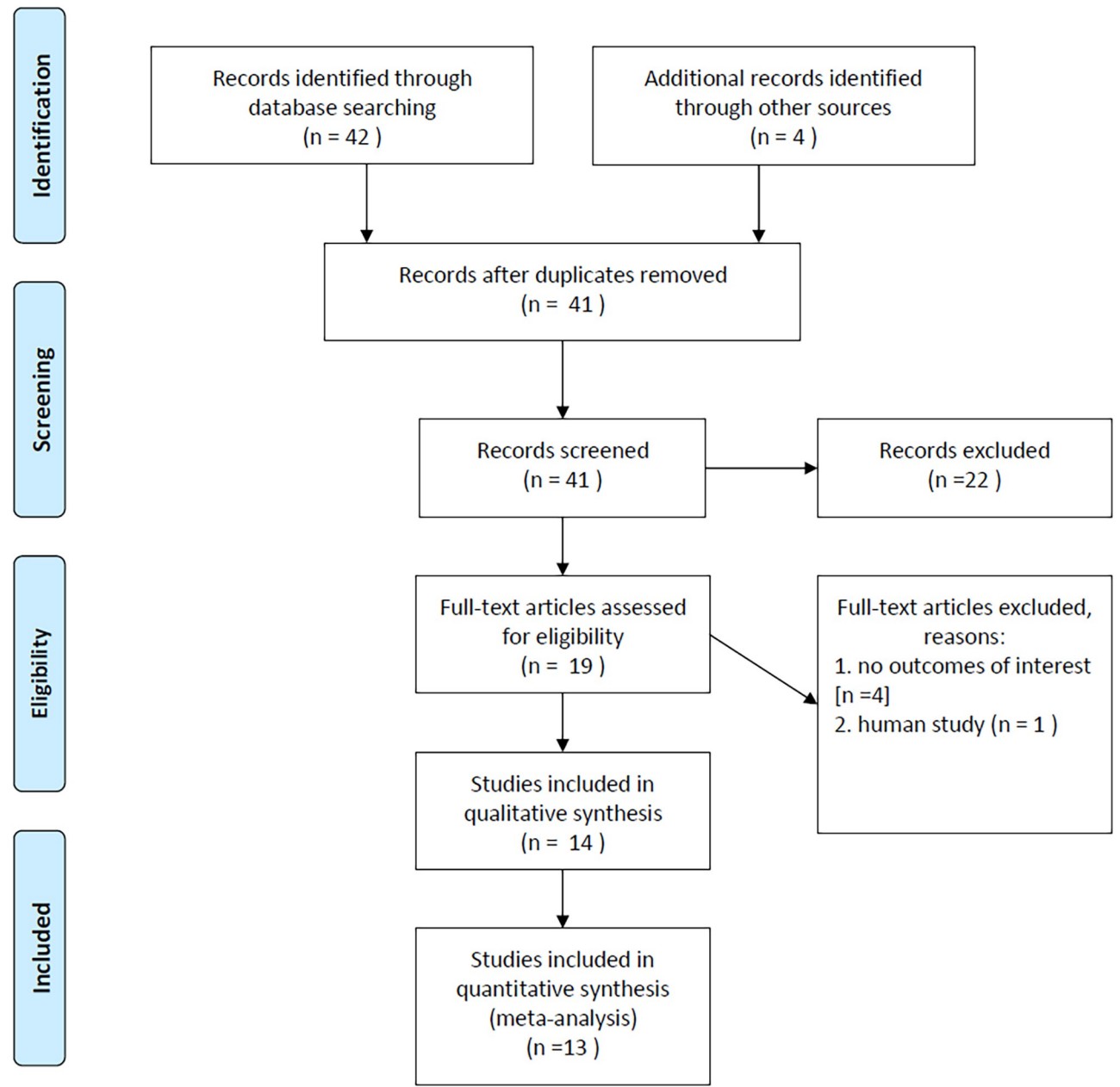

**Fig 1. Flow diagram showing the number of abstracts and articles identified and evaluated during the review.**

When compared with results of using saline as a control, the macroscopic adhesion score was significantly lower in the MB group (SMD, 2.940; 95% CI, 1.344 to 4.537; $I^2$ = 94.0%, Tau = 2.624) (Fig 4). Sensitivity analysis was performed by removing one study at a time; there was no change in the significance of the results (Fig 5).

Subgroup analysis based on surgical procedures showed that there were no macroscopic differences between groups in both laparotomy of the uterine horn (SMD, 2.320; 95% CI, –0.199 to 4.840; $I^2$ = 95.5%, Tau = 2.496) and laparotomy of the cecum or colon (SMD, 2.982; 95% CI, –1.756 to 7.721; $I^2$ = 97.21%, Tau = 4.764).

**Table 1. Characteristics of included studies.**

| First author, publication year | Animal | Surgery | Group | Definition |
|---|---|---|---|---|
| Kluger, 2000 | Female Wistar rats | Laparotomy (uterine horns) | Group 1 | N/S |
| | | | Group 2 | Sham (no induction of adhesions) + 1.00% MB |
| | | | Group 3 | Sham (no induction of adhesions) + N/S |
| | | | Group 4 | 0.13% MB |
| | | | Group 5 | 0.25% MB |
| | | | Group 6 | 0.50% MB |
| | | | Group 7 | 1.00% MB |
| Galili, 1998 | Female Wistar rats | Laparotomy (uterine horns) | Control | N/S |
| | | | MB | 1.0% MB |
| Heydrick, 2007 | Male Wistar rats | Laparotomy | Control | N/S |
| | | | MB | 30 mg/Kg Methylene blue |
| Boztosun, 2012 | Female Wistar rats | Laparotomy (uterine horns) | Control | N/S |
| | | | MB | 1.0% MB |
| El-Sayed, 2016 | Female Wistar rats | Laparotomy (cecum) | Control | N/S |
| | | | MB | 0.525% MB |
| Panahi, 2012 | Female Wistar rats | Laparotomy (cecum) | Control | Sham |
| | | | N/S | N/S |
| | | | MB | 1.0% MB |
| Kalaycı, 2011 | Female Wistar albino rats | Laparotomy | Group 1 | Sham |
| | | | Group 2 | N/S |
| | | | Group 3 | 1.0% MB |
| Cetin, 2004 | Wistar albino rats | Laparotomy (uterine horns) | Sham | No procedure |
| | | | Control | No treatment |
| | | | N/S | N/S |
| | | | MB | 1.0% MB |
| Dinc, 2006 | Male Sprague-Dawley rats | Laparotomy (colon) | Sham | No treatment |
| | | | N/S | N/S |
| | | | MB | 1.0% MB |
| Mahdy, 2008 | Male Wistar rats | Laparotomy (cecum) | Group 1 | 0.5% MB |
| | | | Group 2 | 1.0% MB |
| | | | Group 3 | 5.0% MB |
| | | | Group 4 | 9.0% MB |
| | | | Control | N/S |
| Cetin, 2003 | Female Wistar albino rats | Laparotomy (uterine horns) | Control | Sham |
| | | | N/S | N/S |
| | | | MB | 1.0% MB |
| Yildiz, 2011(1) | Female Sprague-Dawley rats | Laparotomy | Sham | No treatment |
| | | | Control | N/S |
| | | | MB | 1.0% MB |
| Yildiz, 2011(2) | Female Sprague-Dawley rats | Laparotomy | Sham | No treatment |
| | | | Control | N/S |
| | | | MB | 1.0% MB |
| Duran, 2002 | Wistar albino rats | Laparotomy (uterine horns) | Control | No treatment |
| | | | MB | 0.1% MB |

N/S, normal saline; MB, methylene blue

**Table 2. Definition of gross and microscopic adhesion scores.**

| First author, publication year | Gross adhesion score | Microscopic adhesion score |
|---|---|---|
| Kluger, 2000 | Adhesion grades 2, 3, or 4 were considered substantial, while animals with grades 0 or 1 were considered adhesion-free. | Not presented |
| Galili, 1998 | Adhesion grades 2–4 were considered substantial, and animals with adhesion grades 0 or 1 were considered adhesion free. | Not presented |
| Heydrick, 2007 | Adhesion formation was quantified in a blinded fashion with each animal receiving a score based on the percentage of ischemic buttons with fibrinous protoadhesions at 24 h or attached adhesions at 7 d. | Not presented |
| Boztosun, 2012 | The extent of adhesions was graded as follows: 0, no adhesion; 1, 25% of traumatized area; 2, 50% of traumatized area; and 3, total involvement. The severity of adhesions was graded as follows: 0, no resistance to separation; 0.5, some resistance (moderate force required); 1. | Inflammation on the serosal surface, fibroblastic activity, foreign body reaction, collagen formation, and severity of vascular proliferation were semi-quantitatively graded (grade 0 to 4). VEGF, bFGF, PDGF, and TGF- 3 markers were used in immunohistochemical evaluation. Results were scored as 0, 1+, 2+, 3+, and 4+. |
| El-Sayed, 2016 | Extent and type 0: No adhesion; 1: Filmy, transparent, avascular adhesion; 2: Mild, opaque, translucent, avascular adhesion; 3: Moderate, opaque, capillaries present, 4: Severe, opaque, larger vessels. Tenacity 0: No adhesion, 1: Adhesions fall apart, 2: Adhesions lysed with traction, 3: Adhesions sharply dissected, 4: Adhesions not dissectible without damaging organs. | Not presented |
| Panahi, 2012 | Grade 0: No adhesion. Grade 1: The ratio of adhesive area/total treated area in the vermiform processes is. Grade 2: The ratio is 50% and the adhesion is easily dissected. Grade 3: Area of the adhesion is out of consideration; although blunt dissection for the adhesion can be carried out, it is difficult and the intestinal wall will be impaired after the blunt dissection. Grade 4: Area of the adhesion is out of consideration; the adhesion is fast and cannot be bluntly dissected. In addition, there may be adhesion to other organs (liver). | Not presented |
| Kalaycı, 2011 | Cumulative adhesion scoring scale (0), No adhesion; (1), One adhesive band from the omentum to the target organ; (1), One adhesive band from the omentum to the abdominal scar; (1), One adhesive band from the omentum to another place; (1), One adhesive band from the adnexa/epididymal fat bodies to the target organ; (1), One adhesive band from the adnexa/epididymal fat bodies to the abdominal scar; (1), One adhesive band from the adnexa/epididymal fat bodies to another place; (1), Any adhesive band other than described above (e.g., liver to scar); (1), Target organ adherent to the abdominal wall; (1), Target organ adherent to the abdominal scar; (1), Target organ adherent to the bowel; (1), Target organ adherent to the liver or the spleen; (1), Target organ adherent to any other organ. | Not presented |
| Cetin, 2004 | The severity of adhesions was evaluated by a 0- to 5-point scale (0 = no adhesion, 1 = thin film, 2 = thin adhesion, 3 = thick adhesion with focal point, 4 = thick adhesion with planar attachment, and 5 = very thick vascularized adhesion) and the extent of adhesions by a 0- to 4-point scale (0 = no adhesion, 1 = up to 25% of traumatized area, 2 = up to 50% of traumatized area, 3 = up to 75% of traumatized area, 4 = up to 100% of traumatized area). | Not presented |
| Dinc, 2006 | 0: Complete absence of adhesions; 1: Single band of adhesion, between viscera or from viscera to abdominal wall; 2: Two bands, either between viscera or from viscera to abdominal wall; 3: More than 2 bands, between viscera, from viscera to abdominal wall, or whole intestines forming a mass without being adherent to the abdominal wall; and 4: Viscera directly adherent to the abdominal wall, irrespective of number and extent of adhesive bands. | Not presented |

(*Continued*)

**Table 2.** (Continued)

| First author, publication year | Gross adhesion score | Microscopic adhesion score |
|---|---|---|
| Mahdy, 2008 | 0: Complete absence of adhesion; 1: Single band of adhesion, between viscera or from viscera to abdominal wall; 2: Two bands, either between viscera or from viscera to abdominal wall; 3: More than two bands, between viscera, from viscera to abdominal wall, or whole intestines forming a mass without being adherent to the abdominal wall; and 4: Viscera directly adherent to the abdominal wall, irrespective of number and extent of adhesive bands. | Photomicrographs of the adhesions. A, grade 1: A photomicrograph showing a part of an adhered fibrous tissue band with blood vessels and cellular infiltration (H&E x100). B, grade 2: A photomicrograph showing part of two fused fibrous tissue bands with engorged blood vessels (H&E x100). C, grade 3: A photomicrograph showing a part of fibrous tissue bands (H&E x200). D, grade 4: A photomicrograph showing part of a fibrous tissue mass with engorged blood vessels. |
| Cetin, 2003 | Extent 0, no adhesion; 1, up to 25% of traumatized area; 2, between 25% and 50% of traumatized area; and 3, 50%–100% of traumatized area. Severity 0, no resistance to separation; 0.5, some resistance (moderate force required); 1, sharp dissection needed. | Not presented |
| Yildiz, 2011 (1) | 0: Complete absence of adhesion; 1: Single band of adhesion between viscera or from one viscus to the abdominal wall; 2: Two bands, either between viscera or from viscera to the abdominal wall; 3: More than two bands between viscera or from viscera to the abdominal wall; 4: Multiple dense adhesions or viscera directly adherent to the abdominal wall and extent of adhesive bands. | Not presented |
| Yildiz, 2011 (2) | Not presented | The histological sections were examined for the presence and score of adhesion, edema, fibrosis, and mononuclear cell infiltration with a light microscope and photographed. The microscopic score was graded on a scale as follows: (1), mild; (2), moderate; and (3), severe. |
| Duran, 2002 | Adhesion area 0: No adhesion, 1: 25% of surface covered, 2: 50% of surface covered, 3: Completely covered | Not presented |

When compared with the control group, there was no evidence of differences between groups for macroscopic adhesion score (SMD, –0.298; 95% CI, –1.455 to 0.858; $I^2$ = 88.7%, Tau = 1.596) (Fig 6).

There was no change in the significance of the results after performing a sensitivity analysis by removing one study at a time (Fig 7).

Subgroup analysis based on surgical procedures showed that the macroscopic adhesion score was significantly lower in the MB group in laparotomy of the uterine horn (SMD, 0.934; 95% CI, 0.401 to 4.840; $I^2$ = 0.0%, Tau = 0.0); however, there was no evidence of differences between groups in laparotomy of the cecum or colon (SMD, –0.800; 95% CI, –2.364 to 0.763; $I^2$ = 84.58%, Tau = 1.038).

As TSA only supports the analysis of mean difference, two studies that reported different outcome scales (Heydrick 2007 and Panahi 2012) were excluded from TSA. TSA indicated that only 90.9% (329 of 362 patients) of the RIS was accrued. The cumulative *Z* curve (complete blue curve) crossed both the conventional boundary (etched red line) and the sequential monitoring boundary (complete red curve) (S1 Fig in S1 File).

## Regression analysis

Since the results appeared to differ depending on the dose of MB used, the macroscopic adhesion score at different doses of MB was evaluated using meta-regression analysis. Macroscopic adhesion tended to decrease with an increase in the dose of MB (β = –0.350, 95% CI = –0.365 to –0.336, P < 0.001) (Fig 8).

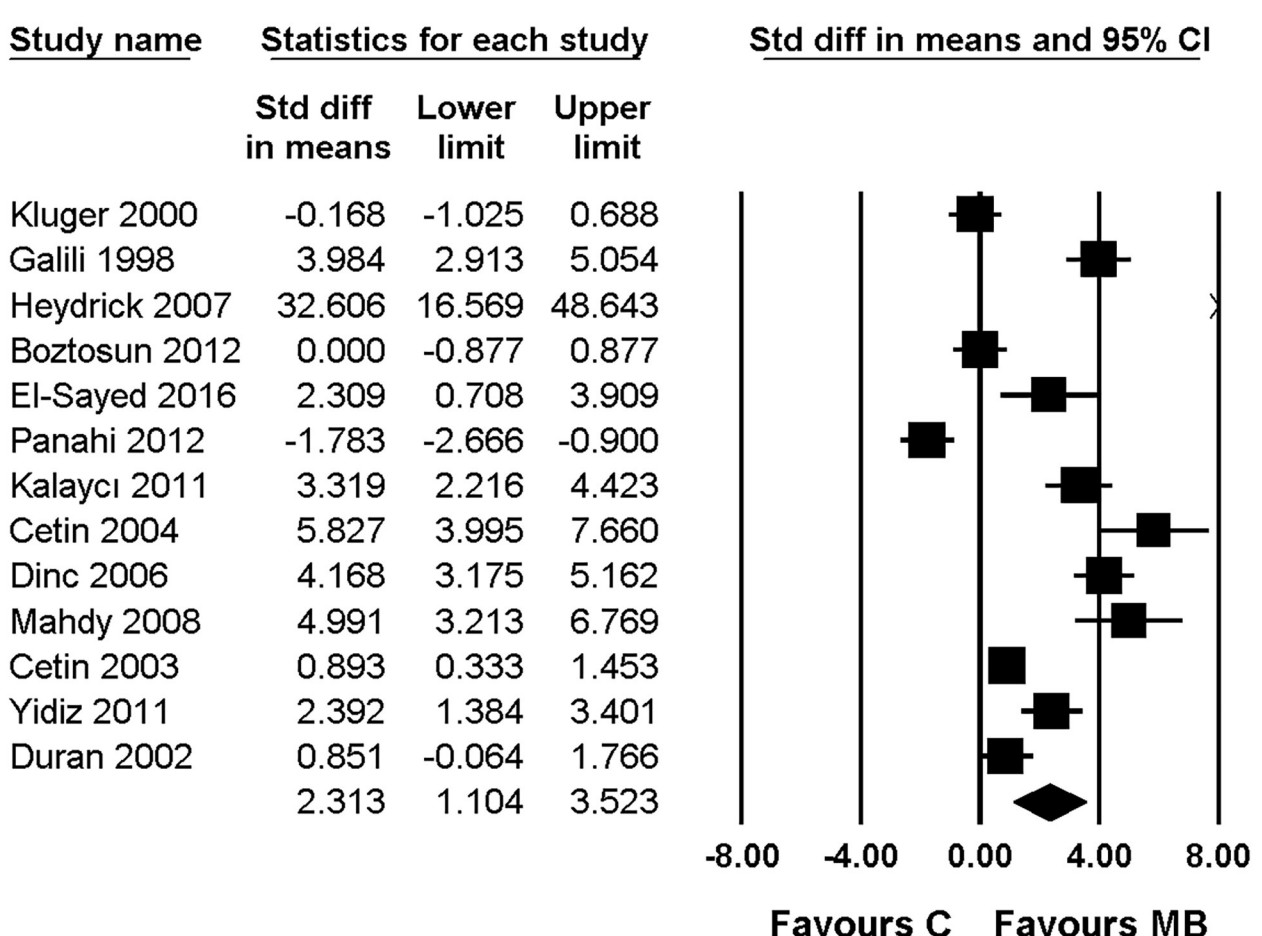

| Study name | Statistics for each study | | | Std diff in means and 95% CI |
|---|---|---|---|---|
| | Std diff in means | Lower limit | Upper limit | |
| Kluger 2000 | -0.168 | -1.025 | 0.688 | |
| Galili 1998 | 3.984 | 2.913 | 5.054 | |
| Heydrick 2007 | 32.606 | 16.569 | 48.643 | |
| Boztosun 2012 | 0.000 | -0.877 | 0.877 | |
| El-Sayed 2016 | 2.309 | 0.708 | 3.909 | |
| Panahi 2012 | -1.783 | -2.666 | -0.900 | |
| Kalaycı 2011 | 3.319 | 2.216 | 4.423 | |
| Cetin 2004 | 5.827 | 3.995 | 7.660 | |
| Dinc 2006 | 4.168 | 3.175 | 5.162 | |
| Mahdy 2008 | 4.991 | 3.213 | 6.769 | |
| Cetin 2003 | 0.893 | 0.333 | 1.453 | |
| Yidiz 2011 | 2.392 | 1.384 | 3.401 | |
| Duran 2002 | 0.851 | -0.064 | 1.766 | |
| | 2.313 | 1.104 | 3.523 | |

Meta Analysis

**Fig 2. Forest plot showing an overall effect of macroscopic adhesion score compared with the combined results of using saline and nothing as control.** The figure depicts individual trials as filled squares with relative sample size and the 95% confidence interval (CI) of the difference as a solid line. The diamond shape indicates the pooled estimate and uncertainty for the combined effect.

### Publication bias

A funnel plot was used for the combined results of using saline and nothing as control and saline only as control, all of which exhibited asymmetrical appearances. The P-values of Egger's test were less than 0.1 when the combined results of using saline and nothing as control (P = 0.031) and saline only as control (P = 0.031) were compared. Thus, we performed trim and fill analysis; however, there was no change in the significance of the results (SMD, 2.155; 95% CI, 0.913 to3.396 vs. SMD, 2.675; 95% CI, 1.046 to4.304) (Figs 9 and 10).

### Microscopic adhesion score

Microscopic adhesion scores were reported in three studies [28, 39, 44].

Mahdy et al. reported that 1.0% MB showed a better anti-adhesive effect than 0.5% MB, 5.0% MB, 9.0% MB, and control in terms of inflammation (0.5 ± 0.4 vs. 1.85 ± 0.3, 2.7 ± 0.4,

## Study name

| Study name | Point | Lower limit | Upper limit | Std diff in means (95% CI) with study removed |
|---|---|---|---|---|
| Kluger 2000 | 2.558 | 1.262 | 3.855 | |
| Galili 1998 | 2.150 | 0.912 | 3.387 | |
| Heydrick 2007 | 2.145 | 0.973 | 3.316 | |
| Boztosun 2012 | 2.546 | 1.241 | 3.850 | |
| El-Sayed 2016 | 2.322 | 1.041 | 3.603 | |
| Panahi 2012 | 2.650 | 1.509 | 3.790 | |
| Kalaycı 2011 | 2.226 | 0.952 | 3.499 | |
| Cetin 2004 | 2.012 | 0.809 | 3.215 | |
| Dinc 2006 | 2.122 | 0.906 | 3.338 | |
| Mahdy 2008 | 2.085 | 0.857 | 3.312 | |
| Cetin 2003 | 2.508 | 1.089 | 3.927 | |
| Yidiz 2011 | 2.324 | 1.014 | 3.635 | |
| Duran 2002 | 2.475 | 1.145 | 3.806 | |
| | 2.313 | 1.104 | 3.523 | |

**Meta Analysis**

**Fig 3. Forest plot showing sensitivity analysis performed by removing one study at a time for an overall effect of macroscopic adhesion score, compared with combined results of using saline and nothing as control.** The figure depicts individual trials as filled squares with relative sample size and the 95% confidence interval (CI) of the difference as a solid line. The diamond shape indicates the pooled estimate and uncertainty for the combined effect.

2.9 ± 0.3, 2.1 ± 0.2, respectively) and fibrosis (0.7 ± 0.2 vs. 1.90 ± 0.4, 3.7 ± 0.4, 3.9 ± 0.3, 2.7 ± 0.4, respectively) [39].

Boztosun et al. reported that 1.0% MB showed lower fibroblastic activity score, vascular endothelial growth factor, platelet derived growth factor, transforming growth factor β, and basic fibroblastic growth factor than the control group (1 [0–2], 0 [0–1], 1 [0–2], 2 [0–3], and 2 [0–4] vs. 2 [2–4], 0.5 [0–4], 2.5 [1–4], 2 [0–4], and 2 [0–4], respectively) [28].

Yildiz et al. reported that MB decreased the adhesion score, edema, fibrosis score, and fibrosis compared with the control group (2.70 ± 0.15, 2.60 ± 0.16, 2.80 ± 0.13, and 2.20 ± 0.13 vs. 0.20 ± 0.13, 1.00 ± 0.21, 0.80 ± 0.13, 0.40 ± 0.16, respectively) [44].

| Study name | Statistics for each study | | | Std diff in means and 95% CI |
|---|---|---|---|---|
| | Std diff in means | Lower limit | Upper limit | |
| Kluger 2000 | -0.168 | -1.025 | 0.688 | |
| Galili 1998 | 3.984 | 2.913 | 5.054 | |
| Heydrick 2007 | 32.606 | 16.569 | 48.643 | |
| Boztosun 2012 | 0.000 | -0.877 | 0.877 | |
| El-Sayed 2016 | 2.309 | 0.708 | 3.909 | |
| Panahi 2012 | -2.896 | -4.151 | -1.642 | |
| Kalaycı 2011 | 3.319 | 2.216 | 4.423 | |
| Cetin 2004 | 5.827 | 3.995 | 7.660 | |
| Dinc 2006 | 7.637 | 5.852 | 9.421 | |
| Mahdy 2008 | 4.991 | 3.213 | 6.769 | |
| Cetin 2003 | 0.827 | 0.181 | 1.473 | |
| Yidiz 2011 | 4.405 | 2.783 | 6.028 | |
| | 2.940 | 1.344 | 4.537 | |

-8.00  -4.00  0.00  4.00  8.00

Favours C    Favours MB

Meta Analysis

**Fig 4. Forest plot showing an overall effect of macroscopic adhesion score compared with results of using saline as control.** The figure depicts individual trials as filled squares with relative sample size and the 95% confidence interval (CI) of the difference as a solid line. The diamond shape indicates the pooled estimate and uncertainty for the combined effect.

## Side effect

None of the studies included in this systematic review and meta-analysis reported any side effects of MB treatment.

**Methodological quality.** A summary of the methodological quality assessment for each study is shown in Table 3. The methodological quality scores ranged from 3 to 5, with two studies scoring 3 or 4 points.

## Discussion

The current systematic review and meta-analysis found that MB has a beneficial effect in preventing postoperative adhesions. The macroscopic adhesion score was significantly lower in the MB group than in the control group, and it tended to decrease as the MB dose increased.

Several studies have been conducted on adhesion after surgery, with abdominal adhesions being a major concern. In a prospective analysis of 210 patients undergoing a laparotomy after having one or more abdominal procedures, 93% had intra-abdominal adhesions as a result of

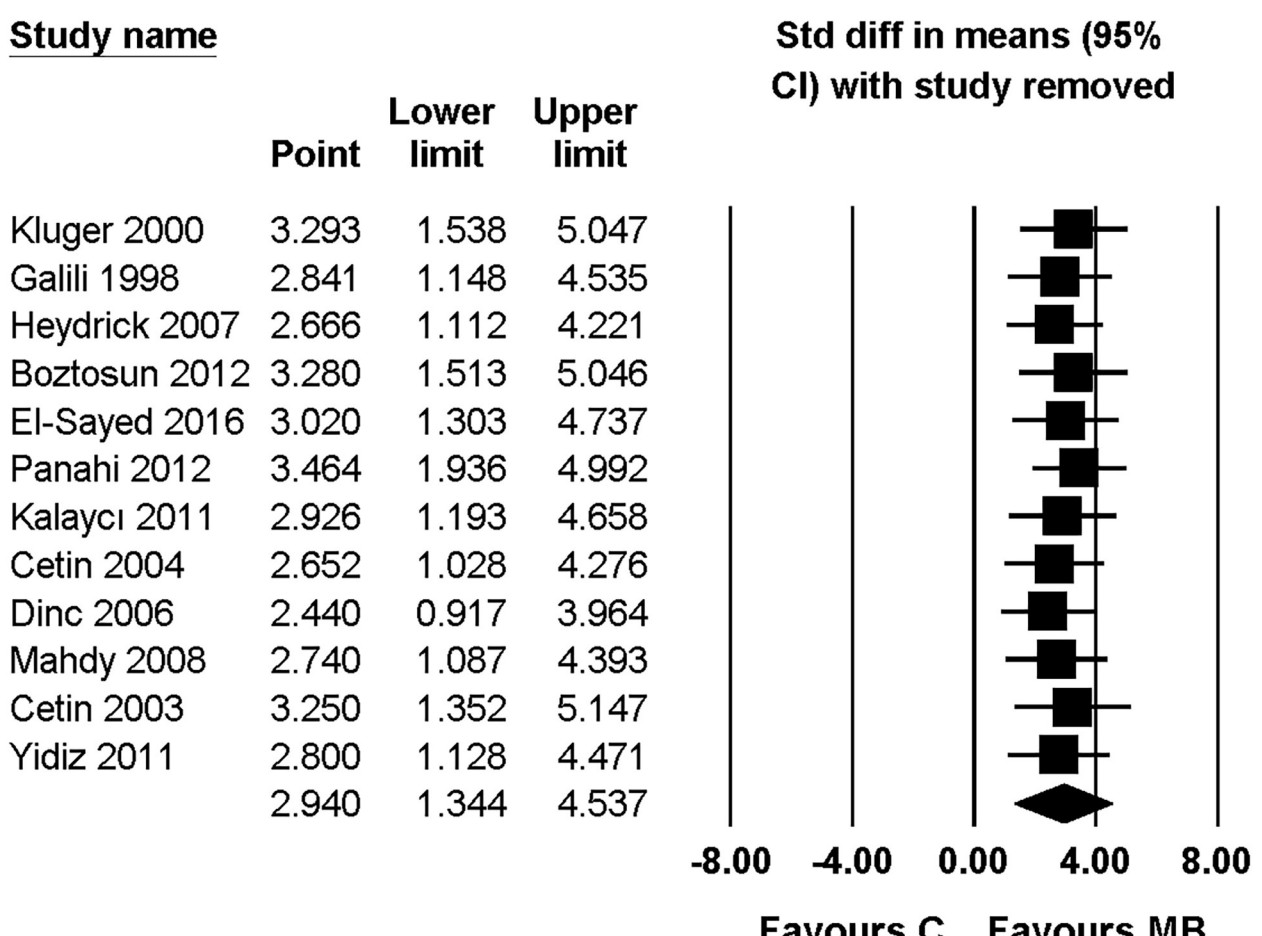

**Meta Analysis**

**Fig 5. Forest plot showing sensitivity analysis performed by removing one study at a time for an overall effect of macroscopic adhesion score, compared with results of using saline as control.** The figure depicts individual trials as filled squares with relative sample size and the 95% confidence interval (CI) of the difference as a solid line. The diamond shape indicates the pooled estimate and uncertainty for the combined effect.

the previous surgery [45]. Postoperative adhesion can cause acute or chronic pain as well as infertility, intestinal obstruction, and reoperation. Furthermore, postoperative adhesion can raise economic problems as it necessitates more treatment, longer hospital stay, and the need for future operations.

To address this issue, numerous studies have been conducted on the prevention of adhesion following surgery. The main strategy for preventing postoperative adhesions is the use of physical and chemical barriers. Physical barriers, which can be characterized as gels, solutions, or films using biomaterials, prevent contact with the surgical site and surrounding tissue. Chemical barriers are anti-adhesive medications that prevent adhesion by inhibiting the adhesion formation pathways. Given the mechanism of adhesion formation following surgery, anti-inflammatory drugs, anticoagulants, antioxidants, or fibrinolytic agents can be viable candidates.

| Study name | Statistics for each study | | | Std diff in means and 95% CI |
|---|---|---|---|---|
| | Std diff in means | Lower limit | Upper limit | |
| Panahi 2012 | 0.000 | -0.877 | 0.877 | |
| Dinc 2006 | -1.595 | -2.455 | -0.736 | |
| Cetin 2003 | 0.977 | 0.321 | 1.633 | |
| Yidiz 2011 | -1.902 | -3.060 | -0.743 | |
| Duran 2002 | 0.851 | -0.064 | 1.766 | |
| | -0.298 | -1.455 | 0.858 | |

-8.00 -4.00 0.00 4.00 8.00

Favours C    Favours MB

Meta Analysis

**Fig 6. Forest plot showing an overall effect of macroscopic adhesion score compared with results of using nothing as control.** The figure depicts individual trials as filled squares with relative sample size and the 95% confidence interval (CI) of the difference as a solid line. The diamond shape indicates the pooled estimate and uncertainty for the combined effect.

MB has been shown to reduce adhesion formation by inhibiting the production of oxygen radicals. According to other investigations, MB inhibits intra-abdominal adhesion development by enhancing peritoneal fibrinolytic activity. Thus, it is essential to summarize and evaluate the current evidence regarding the role of MB as a chemical barrier for the prevention of postoperative adhesion formation by conducting this systematic review and meta-analysis.

Our meta-analysis findings support the anti-adhesive properties of MB. While we performed our meta-analysis using experimental research, some human studies have indicated that MB is clinically available for adhesion prevention. In a retrospective study, Neagoe et al. examined the effectiveness of MB in preventing repeated symptomatic postoperative adhesions in 20 patients who underwent surgeries for intra-abdominal adhesion-related complications and were administered 1% MB [26]. They concluded that using MB during adhesiolysis surgery appears to reduce the recurrence of adhesion-related symptoms. A cohort study of patients undergoing abdominal surgery reported that MB reduced adhesion rates by up to 50% [37]. Consequently, our findings and those of some human studies suggest that MB could be used as a useful agent for the prophylactic treatment of postoperative adhesion in the future. Given the high cost of other postoperative adhesion prevention strategies, MB may be a viable alternative that is easily accessible in clinical practice.

We performed a meta-regression analysis based on MB dose. The most effective concentration must be determined before MB can be clinically evaluated. MB has different effects on abdominal adhesion formation depending on the dose used in experimental studies, and there

**Study name**

| | Point | Lower limit | Upper limit |
|---|---|---|---|
| Panahi 2012 | -0.384 | -1.881 | 1.113 |
| Dinc 2006 | 0.045 | -1.083 | 1.174 |
| Cetin 2003 | -0.640 | -1.899 | 0.619 |
| Yidiz 2011 | 0.068 | -1.096 | 1.233 |
| Duran 2002 | -0.590 | -1.976 | 0.796 |
| | -0.298 | -1.455 | 0.858 |

Meta Analysis

**Fig 7. Forest plot showing sensitivity analysis performed by removing one study at a time for an overall effect of macroscopic adhesion score, compared with results of using nothing as control.** The figure depicts individual trials as filled squares with relative sample size and the 95% confidence interval (CI) of the difference as a solid line. The diamond shape indicates the pooled estimate and uncertainty for the combined effect.

are conflicting reports on the dose-dependent anti-adhesive effects of MB. While MB inhibited adhesions at 1% concentration, it promoted adhesions at higher or lower concentrations, according to Mahdy et al.'s findings [39]. Galili et al. found that injecting MB intraperitoneally decreased the incidence and severity of peritoneal adhesions [41]. Prien et al. discovered that when mice were administered 9% MB intraperitoneally, abdominal adhesion was formed, most likely due to macrophage activation [46]. In addition, high concentrations of MB have been linked to side effects such as pericardial pain, dyspnea, restlessness, and tremor [47]. Although many studies have indicated different optimal concentrations and volumes of MB, our meta-regression analysis revealed that macroscopic adhesion decreased as the dose of MB increased. We also conducted a TSA on adhesion score results to determine whether our findings could be considered as firm evidence. The evidence presented in this study was sufficient to support the use of MB to reduce postoperative adhesions. Further research into the optimal methylene dose and volume, as well as the unfavorable effects on other outcomes in the human population, is required.

This study has several limitations. First, the results of the meta-analysis revealed substantial heterogeneity. Included studies were conducted under diverse protocols under varying concentrations of MB and different types of surgery, which can lead to considerable heterogeneity. We conducted a subgroup analysis by dividing the control groups based on whether they were

## Regression of MB on Mean

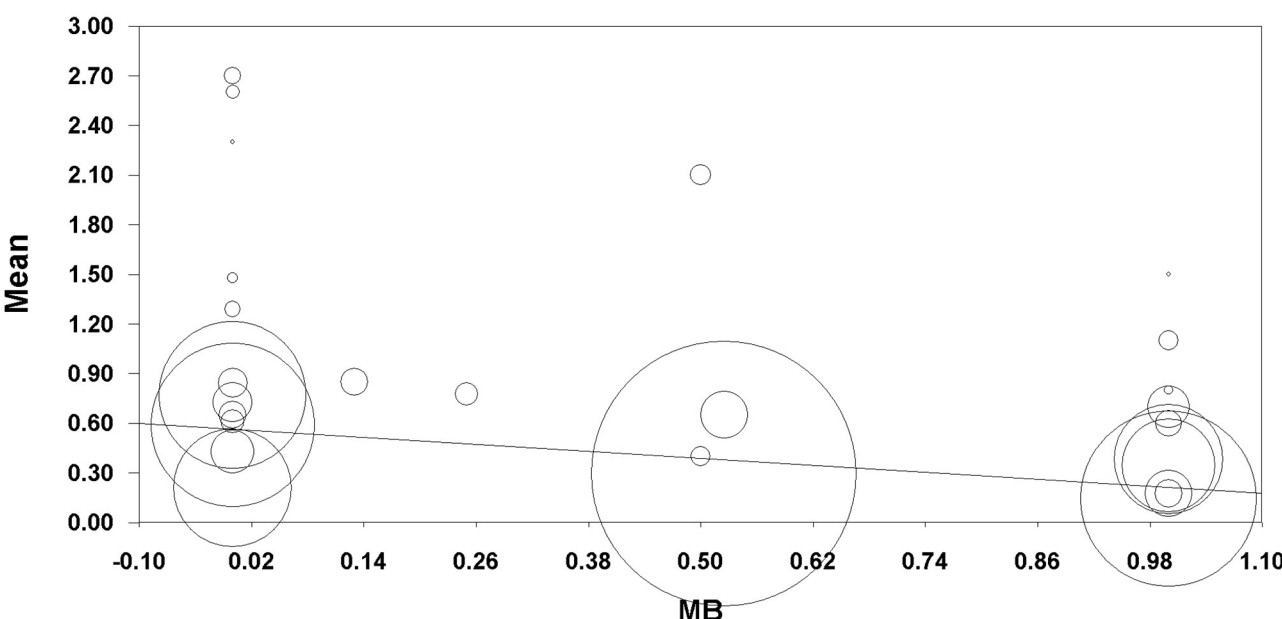

**Fig 8. Meta-regression of mean macroscopic adhesion score by a dose of methylene blue.** The X-axis represents the dose of methylene blue and Y-axis represents the macroscopic adhesion score. The size of the data marker is proportional to the weight in the meta regression.

## Funnel Plot of Precision by Std diff in means

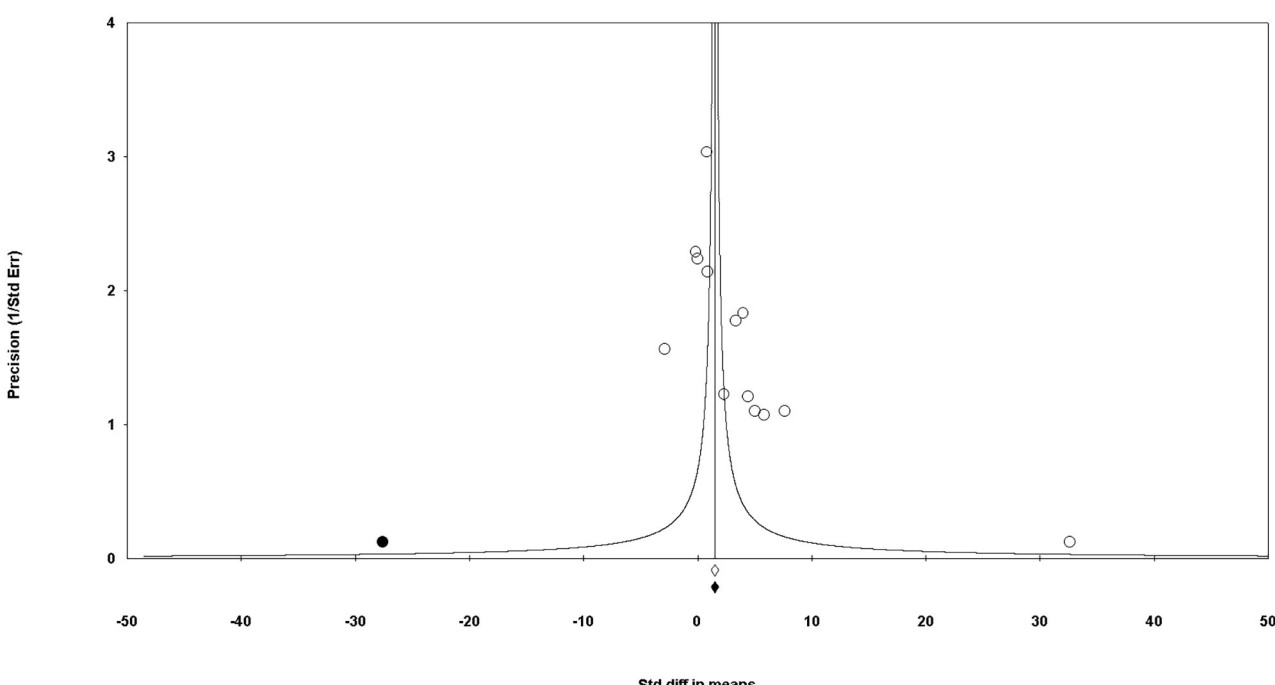

**Fig 9. Funnel plot of comparison: Methylene blue compared with combined results of using saline and nothing as control; outcome—macroscopic adhesion score.** White circles: included comparisons. Black circles: imputed comparisons using the trim-and-fill method. White diamond: pooled observed log risk ratio. Black diamond: pooled imputed log risk ratio.

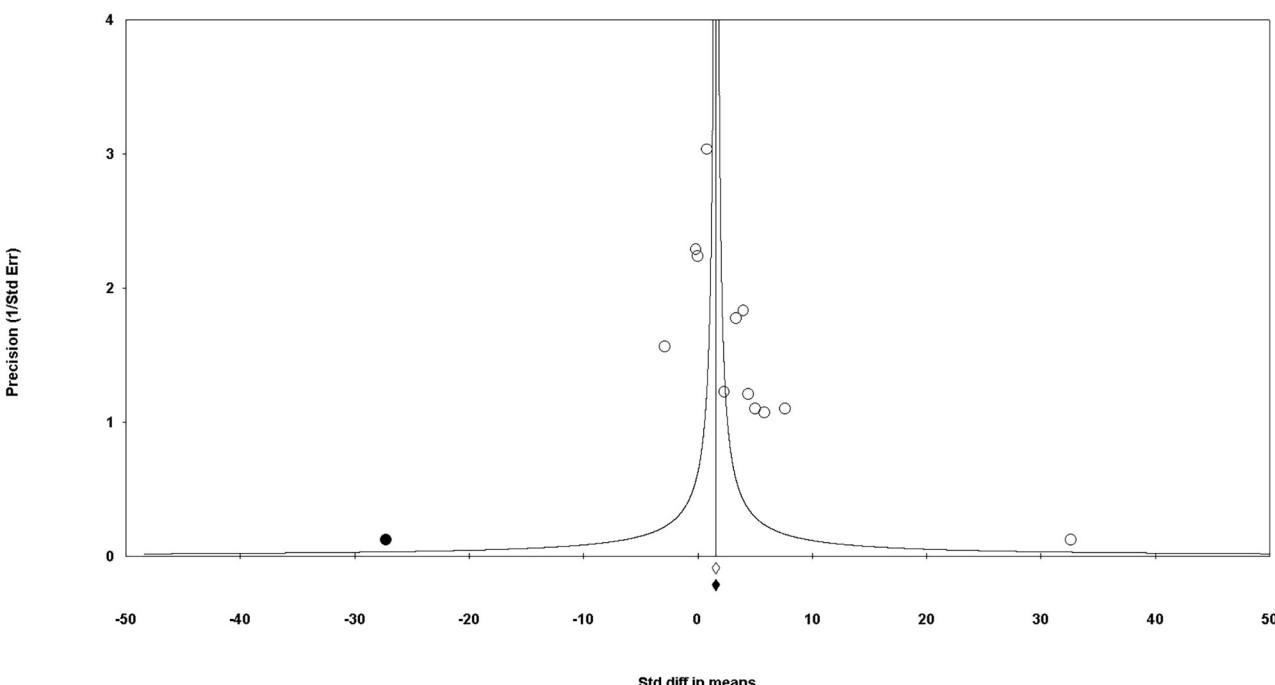

**Fig 10. Funnel plot of comparison: Methylene blue compared with saline used as control; outcome—macroscopic adhesion score.** White circles: included comparisons. Black circles: imputed comparisons using the trim-and-fill method. White diamond: pooled observed log risk ratio. Black diamond: pooled imputed log risk ratio.

**Table 3. Assessment of methodological quality.**

| First author, publication year | Statement of random allocation | Husbandry conditions | Compliance with animal welfare regulations | Peer reviewed | Potential conflict of interest | Score |
|---|---|---|---|---|---|---|
| Kluger, 2000 | 1 | 1 | 1 | 1 | 1 | 5 |
| Galili, 1998 | 1 | 1 | 1 | 1 | 1 | 5 |
| Heydrick, 2007 | 1 | 1 | 1 | 1 | 1 | 5 |
| Boztosun, 2012 | 1 | 1 | 1 | 1 | 1 | 5 |
| El-Sayed, 2016 | 1 | 1 | 1 | 1 | 1 | 5 |
| Panahi, 2012 | 1 | 1 | 1 | 1 | 1 | 5 |
| Kalaycı, 2011 | 0 | 1 | 1 | 1 | 1 | 4 |
| Cetin, 2004 | 1 | 1 | 1 | 1 | 1 | 5 |
| Dinc, 2006 | 1 | 1 | 1 | 1 | 1 | 5 |
| Mahdy, 2008 | 1 | 1 | 1 | 1 | 1 | 5 |
| Cetin, 2003 | 1 | 1 | 1 | 1 | 1 | 5 |
| Yildiz, 2011 (1) | 1 | 1 | 1 | 1 | 1 | 5 |
| Yildiz, 2011 (2) | 1 | 1 | 1 | 1 | 1 | 5 |
| Duran, 2002 | 0 | 0 | 1 | 1 | 1 | 3 |

Methodological quality was assessed based on statements of 1) random allocation into treatment and control groups, 2) husbandry conditions (e.g., light/dark cycle, temperature, access to water, and environmental enrichment), 3) compliance with animal welfare regulations, and 4) potential conflicts of interests, and whether the study appeared in a peer-reviewed publication. Each article was assessed independently by two reviewers and scored on a scale of 0 to 5 points.

given saline or nothing and based on the surgical procedure, and we also performed sensitivity analyses on all included outcomes. Furthermore, we conducted a meta-regression of the MB dose applied, and examined the relationship between the anti-adhesive effect and MB dose, considering the differences in MB dose across trials. TSA was performed to address the issue of limited study numbers, and the results suggest that the evidence from the current analysis is sufficient to propose the anti-adhesive effect of MB in a preclinical study. Finally, as the studies included were experimental, more recent evidence from human trials on MB is needed for clinical application. As evidence of a preclinical investigation, the current findings from our study can serve as a basis for clinical trials. Despite these limitations, our study demonstrated strength by implementing a rigorous methodology to provide the first systematic review and meta-analysis evaluating the anti-adhesive effect of MB in preventing postoperative adhesion.

In conclusion, MB showed a beneficial effect on intraperitoneal adhesion after laparotomy, and adhesion was reduced as the dose of MB was increased. The evidence from this study appears to be sufficient to reach a definitive conclusion, indicating the possibility of the clinical application of MB as a useful chemical barrier for the prevention of postoperative adhesion.

## Supporting information

**S1 Checklist. PRISMA 2020 checklist.**
(DOCX)

**S1 File.**
(DOCX)

**S2 File.**
(XLSX)

## Author Contributions

**Conceptualization:** Su Hyun Seo, Geun Joo Choi, Oh Haeng Lee, Hyun Kang.

**Data curation:** Su Hyun Seo, Geun Joo Choi, Hyun Kang.

**Formal analysis:** Su Hyun Seo, Geun Joo Choi, Oh Haeng Lee, Hyun Kang.

**Investigation:** Su Hyun Seo, Geun Joo Choi.

**Methodology:** Su Hyun Seo, Oh Haeng Lee, Hyun Kang.

**Project administration:** Oh Haeng Lee, Hyun Kang.

**Resources:** Oh Haeng Lee, Hyun Kang.

**Software:** Geun Joo Choi, Oh Haeng Lee, Hyun Kang.

**Supervision:** Geun Joo Choi.

**Validation:** Su Hyun Seo, Geun Joo Choi.

**Visualization:** Geun Joo Choi.

**Writing – original draft:** Su Hyun Seo, Geun Joo Choi, Oh Haeng Lee, Hyun Kang.

**Writing – review & editing:** Hyun Kang.

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
