## [Decision Letter · Decision Letter 0]

17 Feb 2022

PONE-D-21-39003Effect of methylene blue on experimental postoperative adhesion: a systematic review and meta-analysisPLOS ONE

DearKang,

Thank you for submitting your manuscript to PLOS ONE. After careful consideration, we feel that it has merit but does not fully meet PLOS ONE’s publication criteria as it currently stands. Therefore, we invite you to submit a revised version of the manuscript that addresses the points raised during the review process.

Please submit your revised manuscript within Apr 03 2022 11:59PM. If you will need more time than this to complete your revisions, please reply to this message or contact the journal office at plosone@plos.org. Please include the following items when submitting your revised manuscript:A rebuttal letter that responds to each point raised by the academic editor and reviewer(s). You should upload this letter as a separate file labeled 'Response to Reviewers'.A marked-up copy of your manuscript that highlights changes made to the original version. You should upload this as a separate file labeled 'Revised Manuscript with Track Changes'.An unmarked version of your revised paper without tracked changes. You should upload this as a separate file labeled 'Manuscript'.

We look forward to receiving your revised manuscript.

Kind regards,

Yuan-Pin Hsu

Academic Editor

PLOS ONE

Journal Requirements:

Reviewers' comments:

Reviewer's Responses to Questions

**Comments to the Author**

1. Is the manuscript technically sound, and do the data support the conclusions?

Reviewer #1: Yes

Reviewer #2: Yes

Reviewer #3: Yes

2. Has the statistical analysis been performed appropriately and rigorously? 

Reviewer #1: Yes

Reviewer #2: Yes

Reviewer #3: Yes

3. Have the authors made all data underlying the findings in their manuscript fully available?

Reviewer #1: Yes

Reviewer #2: Yes

Reviewer #3: Yes

4. Is the manuscript presented in an intelligible fashion and written in standard English?

Reviewer #1: Yes

Reviewer #2: Yes

Reviewer #3: Yes

5. Review Comments to the Author

Reviewer #1: Summary

The authors have performed a systematic review and meta-analysis of the literature on the effects of methylene blue on post-operative adhesions in animal models.

Comments

This work is original and informative. While meta-analyses can be found on human studies, I was not able to find the equivalent in experimental models.

It would possibly add to the value of the study if the authors expanded on any reported side effects of the methylene blue treatment from the selected manuscripts.

While the manuscript is presented in an intelligible fashion, there are some grammatical errors.

Reviewer #2: the authors are presenting the first meta-analysis for the effect of Methylene Blue on post-operative intra-abdominal adhesions. The authors implemented that no language exclusion was applied. I was wondering if in the case of a manuscript that is written in Spanish or other language that the authors are not familiar with, what was the situation and if that did not happen, I believe it should be mentioned in the methods section.

Some repetitive sentences were encountered when explaining the method used to resolve conflict between the two authors responsible for the literature search / review. It would be of significance if these sentences were altered.

Some minor grammatical errors were found that can be easily fixed with a simple revision.

Reviewer #3: The author, in the limitation, mentioned that there is significant heterogeneity in the studies due to the different surgical approaches. Could performing a subgroup analysis based on surgical procedures be helpful to the readers?

The adhesion grading shown in table 2 is not uniform among the studies. How did the author attempt to unify the values to calculate a pooled estimate?

6. PLOS authors have the option to publish the peer review history of their article (what does this mean?). If published, this will include your full peer review and any attached files.

Reviewer #1: No

Reviewer #2: No

Reviewer #3: No

---

## [Author Response · Author response to Decision Letter 0]

17 Mar 2022

Dear Yuan-Pin Hsu Academic Editor, PLOS ONE,

We thank sincerely Editor and Reviewers of the ‘PLOS ONE’ for taking their precious time to review our paper. Your constructive, meticulous and considerate comments were great guidance for our aforementioned manuscript. According to your precious comments and suggestions, we sincerely and earnestly tried to response for your letter. We want to express my heartfelt gratitude for your comments once more. 

We have made some corrections in the manuscript after going over your comments. We highlighted the modification made to the original document by using red colored text. The changes are summarized below:

Reviewer #1: Summary

The authors have performed a systematic review and meta-analysis of the literature on the effects of methylene blue on post-operative adhesions in animal models.

Comments

This work is original and informative. While meta-analyses can be found on human studies, I was not able to find the equivalent in experimental models.

It would possibly add to the value of the study if the authors expanded on any reported side effects of the methylene blue treatment from the selected manuscripts.

Our response: Thank you for reviewer’s support and suggestion. We agree with reviewer’s comment. According to reviewer’s comment, we reviewed all studies included in this systematic review and meta-analysis for the side effects of methylene blue. However, all the studies included did not report any side effects of methylene blue. We describe it in the manuscript (Methods section 1st paragraph on page 8, Results section 2nd paragraph on page 25)

While the manuscript is presented in an intelligible fashion, there are some grammatical errors.

Our response: Thank you for reviewer’s comment. According to reviewer’s comment, we.re-performed English editing from editage and Job code is ACUNE_7542_2.

Reviewer #2: the authors are presenting the first meta-analysis for the effect of Methylene Blue on post-operative intra-abdominal adhesions. The authors implemented that no language exclusion was applied. I was wondering if in the case of a manuscript that is written in Spanish or other language that the authors are not familiar with, what was the situation and if that did not happen, I believe it should be mentioned in the methods section.

Our response: Thank you for reviewer’s comment. We planned to consult and co-work with experts affiliated with our university, when foreign language translation was necessary. 

In the first stage of study selection, namely when study selection was performed from the title or abstract, full text of some articles were written in many other languages. But title and abstract of these articles are written in English. Therefore, we can perform first stage of study selection. 

And in the second stage of study selection, namely when full text versions were evaluated, all the articles were written in English. Therefore, we did not consult and co-work with experts affiliated. 

According to reviewer’s comment, we described this in the manuscript. (Methods section 1st paragraph on page 6, Results section 1st paragraph on page 11)

Some repetitive sentences were encountered when explaining the method used to resolve conflict between the two authors responsible for the literature search / review. It would be of significance if these sentences were altered.

Our response: Thank you for reviewer’s comment. According to reviewer’s comment, we altered these sentences. (Methods section last paragraph on page 6,)

Some minor grammatical errors were found that can be easily fixed with a simple revision.

Our response: Thank you for reviewer’s comment. According to reviewer’s comment, we.re-performed English editing from editage and Job code is ACUNE_7542_2.

Reviewer #3: The author, in the limitation, mentioned that there is significant heterogeneity in the studies due to the different surgical approaches. Could performing a subgroup analysis based on surgical procedures be helpful to the readers?

Our response: Thank you for reviewer’s suggestion for the betterment of the manuscript. According to reviewer’s comment, we performed subgroup analysis based on surgical procedures. However, the heterogeneity was not decreased. We added the description for subgroup analysis based on surgical procedures in the manuscript. (Methods section last paragraph on page 8, Results section 2nd, 3rd paragraph on page 22, 1st paragraph on page 23, Discussion section 2nd paragraph on page 30)

The adhesion grading shown in table 2 is not uniform among the studies. How did the author attempt to unify the values to calculate a pooled estimate?

Our response: Thank you for reviewer’s comment. As reviewer recommended, the results in table 2 is not uniform among the studies. As described in results section on page 13, studies used 5 point scale, 4 point scale, 5 and 6 point scale, cumulative scale or percentage. All these scales showed the severity of macroscopic adhesion score. Therefore, as described in the manuscript, we used the standardized mean difference to calculate a pooled estimate. The standardized mean difference is commonly used as a summary statistic in meta-analysis when the studies all assess the same outcome but measure it in a variety of ways

We hope the revised manuscript will better meet the requirements of the ‘PLOS ONE’ for publication. Again, we are most grateful for the constructive review by Editor and reviewers of the ‘PLOS ONE’.

Sincerely yours,

Hyun Kang, MD, PhD, MPH 

Professor,

Department of Anesthesiology and Pain Medicine, 

Chung-Ang University College of Medicine,

84 Heukseok-ro, Dongjak-gu, Seoul, 06911, Republic of Korea.

Tel: +82-2-6299-2571, 2579, 2586; Fax: +82-2-6299-2585

E-mail: roman00@naver.com

---

## [Decision Letter · Decision Letter 1]

25 Apr 2022

Effect of methylene blue on experimental postoperative adhesion: a systematic review and meta-analysis

PONE-D-21-39003R1

Dear Dr. Kang,

We’re pleased to inform you that your manuscript has been judged scientifically suitable for publication and will be formally accepted for publication once it meets all outstanding technical requirements.

Kind regards,

Yuan-Pin Hsu

Academic Editor

PLOS ONE

Additional Editor Comments (optional):

Reviewers' comments:

Reviewer's Responses to Questions

**Comments to the Author**

1. If the authors have adequately addressed your comments raised in a previous round of review and you feel that this manuscript is now acceptable for publication, you may indicate that here to bypass the “Comments to the Author” section, enter your conflict of interest statement in the “Confidential to Editor” section, and submit your "Accept" recommendation.

Reviewer #1: All comments have been addressed

Reviewer #2: All comments have been addressed

Reviewer #3: All comments have been addressed

2. Is the manuscript technically sound, and do the data support the conclusions?

Reviewer #1: Yes

Reviewer #2: Yes

Reviewer #3: Yes

3. Has the statistical analysis been performed appropriately and rigorously? 

Reviewer #1: Yes

Reviewer #2: Yes

Reviewer #3: Yes

4. Have the authors made all data underlying the findings in their manuscript fully available?

Reviewer #1: Yes

Reviewer #2: Yes

Reviewer #3: Yes

5. Is the manuscript presented in an intelligible fashion and written in standard English?

Reviewer #1: Yes

Reviewer #2: Yes

Reviewer #3: Yes

6. Review Comments to the Author

Reviewer #1: Thank you for addressing my concerns.

Reviewer #2: The authors addressed all my comments. This is a very well constructed manuscript and it is presenting a novel idea.

Reviewer #3: (No Response)

7. PLOS authors have the option to publish the peer review history of their article (what does this mean?). If published, this will include your full peer review and any attached files.

Reviewer #1: No

Reviewer #2: No

Reviewer #3: No

---

## [Editor Report · Acceptance letter]

10 May 2022

PONE-D-21-39003R1 

Effect of methylene blue on experimental postoperative adhesion: a systematic review and meta-analysis 

Dear Dr. Kang:

I'm pleased to inform you that your manuscript has been deemed suitable for publication in PLOS ONE. Congratulations! Your manuscript is now with our production department. 

Kind regards, 

on behalf of

Dr. Yuan-Pin Hsu 

Academic Editor

PLOS ONE